# RNAseq analysis of oocyte maturation from the germinal vesicle stage to metaphase II in pig and human

Feng Tang, Katja Hummitzsch, Raymond J. Rodgers👤*

School of Biomedicine, Robinson Research Institute, The University of Adelaide, Adelaide, SA, Australia

* ray.rodgers@adelaide.edu.au

## Abstract

During maturation oocytes at the germinal vesicle (GV) stage progress to metaphase II (MII). However, during in vitro maturation a proportion often fail to progress. To understand these processes, we employed RNA sequencing to examine the transcriptome profile of these three groups of oocytes from the pig. We compared our findings with similar public oocyte data from humans. The transcriptomes in oocytes that failed to progress was similar to those that did. We found in both species, the most upregulated genes in MII oocytes were associated with chromosome segregation and cell cycle processes, while the most down regulated genes were relevant to ribosomal and mitochondrial pathways. Moreover, those genes involved in chromosome segregation during GV to MII transition were conserved in pig and human. We also compared MII and GV oocyte transcriptomes at the isoform transcript level in both species. Several thousands of genes (including *DTNBP1*, *MAPK1*, *RAB35*, *GOLGA7*, *ATP1A1* and *ATP2B1*) identified as not different in expression at a gene transcript level were found to have differences in isoform transcript levels. Many of these genes were involved in ATPase-dependent or GTPase-dependent intracellular transport in pig and human, respectively. In conclusion, our study suggests the failure to progress to MII in vitro may not be regulated at the level of the genome and that many genes are differentially regulated at the isoform level, particular those involved ATPase- or GTPase-dependent intracellular transport.

## Introduction

During prenatal life, immature oocytes in the developing ovary begin meiotic division, but then arrest as primary oocytes at prophase I of meiosis I until puberty. This meiotic stage is characterized by an oocyte with a large nucleus (germinal vesicle; GV) and visible nucleolus. Each primary oocyte is surrounded by pre-granulosa cells and together they constitute a primordial follicle. Daily, a small proportion of primordial follicles are activated and start their development via primary follicles, preantral follicles to antral follicles. The latter are so named as they contain a follicular-fluid-filled antrum. The antrum is bounded by multiple layers of granulosa cells and two specialized stromal layers, the theca interna and theca externa. During

**Data Availability Statement:** All relevant data have been uploaded as supporting documents. There are 22 supplementary tables and 5 supplementary figures. The Tables are excel files so that they can be easily further analyzed by any reader wishing to

examine the data. Raw sequencing data, images, and processed data have been deposited in the Gene Expression Omnibus (GEO) repository under accession GEO: GSE237026. All original code has been deposited at GitHub: https://github.com/tf1993614/Pig-Human-RNAseq-analysis/tree/main. The public data used are cited as 'Public human oocytes FASTQ sequencing files were downloaded from the Gene Expression Omnibus (GEO) database using the accession number GSE164371.

**Funding:** F.T. was supported by Adelaide University (https://www.adelaide.edu.au/) China Fee Scholarship and the Faculty of Sciences, Engineering and Technology, University if Adelaide. K.H. was supported by The University of Adelaide Robinson Research Institute Career Development Fellowship and a Building On Ideas Grant. The funders played no role in the study design, data collection and analysis, decision to publish, or preparation of the manuscript.

**Competing interests:** The authors have declared that no competing interests exist.

follicular growth the oocyte enlarges considerably, develops a zona pellucida and interacts with the surrounding somatic cumulus cells via gap junctions. At this stage the oocyte is prevented from progressing further into meiosis by high levels of cyclic adenosine monophosphate (cAMP)/cyclic guanosine monophosphate (cGMP) transported from surrounding cumulus cells into the oocyte via gap junctions [1, 2]. During the reproductive cycle the luteinizing hormone (LH) surge initiates the process of ovulation of oocytes from large antral follicles and the gap junctions between the cumulus cells and the oocyte break down, significantly reducing the cAMP/cGMP concentration in the oocyte [2]. This initiates meiosis resumption and this progresses to metaphase II (MII) where meiosis stops again. The process of developing from a GV oocyte to a MII oocyte is called oocyte maturation. The MII oocyte will complete meiosis II if fertilisation by sperm occurs.

During oocyte maturation, a series of biological events have to occur to secure accurate chromosome segregation and cell division resulting in a healthy MII oocyte with fertilisation competence. Interruption of spindle formation, which is the major mediator of chromosome segregation, can lead to meiotic arrest and impairment of the oocyte maturation rate [3, 4]. Moreover, spindle microtubules attachment to chromosomes has to occur in an amphitelic way, meaning the two sister kinetochores should attach from opposite spindle poles. Otherwise, it would result in incorrect chromosome segregation, and the production of aneuploid (abnormal number of chromosomes) oocytes [5, 6] and subsequent miscarriages or birth defects if fertilised. In addition to the important function of cytoskeleton, oxidative phosphorylation (OXPHOS) also plays an essential role in the regulation of energy-intensive oocyte maturation. A positive correlation between oocyte developmental potential and its ATP content has been confirmed [7]. Oocyte aneuploidy can happen due to mitochondrial dysfunction and a decrease in mitochondria-derived ATP [8, 9]. Reactive oxygen species (ROS) are an unavoidable by-product of the electron transport chain in mitochondria during OXPHOS. Studies have shown that ROS might cause aneuploidy by premature loss of cohesion and chromosome segregation errors [10]. Therefore, maintaining a redox balance puts great constraint on oocyte maturation [11]. Aneuploidy events are more prevalent during meiosis I than meiosis II, which leads to the question if the oxidative response is different resulting in the oocyte during meiosis I being more susceptible to ROS-damage.

Although the underlying transcriptomic/proteomic differences between GV and MII oocytes have been studied in different species [12–14], to the best of our knowledge no transcriptomic profiles with reliable replicates ($\geq$5) are established, especially in large mammals such as the pig. Additionally, most oocyte-relevant RNA sequencing data has only been analysed at the gene level in the past. However, with the release of novel analysing software like *Salmon* and *Kallisto* [15, 16], analysing RNA sequencing (RNAseq) data at isoform transcript resolution is no longer a problem and has the potential to reveal significant biological clues otherwise previously missed.

In this study, we collected oocytes at three major developmental stages including GV, MII and damaged (referring to oocytes that did not progress to producing the first polar body after *in vitro* maturation) from pig ovaries to perform RNAseq and identified changes in genes/transcripts required for oocytes to transition from GV to MII stage and main upstream regulators and pathways involved in oocyte maturation. Our focus lay on gene/transcript level alterations regarding oxidative stress response/ antioxidants, oxidative phosphorylation and mitochondria, endoplasmic reticulum (ER) stress, oocyte developmental marker, cell cycle, ribosomal, and cytoskeleton/ extracellular matrix. To identify differences in genes and transcripts, which could lead to a decline in oocyte competence by delay or breakdown of development, we analysed MII oocytes with a polar body and 'damaged' oocytes without a polar body following *in vitro* incubation. We also compared our transcriptomic profiles for GV and MII

pig oocytes with corresponding human oocyte stages using public human oocyte RNAseq datasets.

## Materials

### Oocyte collection and in vitro maturation

Sow ovaries were obtained from a local abattoir and transported to the laboratory in warm 0.9% sodium chloride solution (Baxter, Old Toongabbie, NSW, Australia) at 37°C. Follicular fluid including cumulus oocyte complexes (COCs) was then collected from follicles with a size of 5 mm using a vacuum pump. COCs with a uniform cytoplasm and several layers of cumulus cells were separated from the follicular fluid by mouth-pipetting, followed by washing with phosphate buffered saline (PBS) to remove cell debris. For the collection of GV stage oocytes, we removed cumulus cells by digestion of a subset of fresh COCs with 5 μg/mL hyaluronidase (Sigma-Aldrich/Merck, St. Louis, MO, USA), followed by 5 mins vortex at RT. Then, the remaining COCs were cultured for 42-44h in Medium 199 (GIBCO/Life Technologies) supplemented with 100 μg/ml sodium pyruvate (Sigma-Aldrich/Merck, St. Louis, MO, USA), 75μg/ml penicillin (Sigma-Aldrich/Merck, St. Louis, MO, USA), 50 μg/ml streptomycin (Sigma-Aldrich/Merck, St. Louis, MO, USA), 1 mM cysteamine (Sigma-Aldrich/Merck, St. Louis, MO, USA), 5 μg/ml insulin (Sigma-Aldrich/Merck, St. Louis, MO, USA), 10 IU/ml pregnant mare serum gonadotrophin (PMSG; Intervet Schering-Plough Animal Health), 10 IU/ml human choriongonadotrophin (HCG; Intervet Schering-Plough Animal Health), $10^{-2}$ μg/ml EGF (Sigma-Aldrich/Merck, St. Louis, MO, USA) and 1 ml sow follicular fluid at 38.5°C and 5% $CO_2$ to induce maturation from GV stage to MII stage. After *in vitro* maturation for 42-44h, the COCs were treated with 5 μg/ml hyaluronidase (Sigma-Aldrich/Merck, St. Louis, MO, USA) to remove cumulus cells and then observed under a stereomicroscope to determine the presence of a first polar body. Oocytes with a first polar body were classified as 'MII group', whereas oocytes lacking a first polar body were classified as 'damaged group' (S1 Fig). All denuded oocytes were washed by calcium/magnesium-free Dulbecco's phosphate buffered saline (PBS) buffer (GIBCO/Life Technologies) several times and immediately stored at -80°C until preparation for RNAseq. We performed 5 biologically independent collections, whereby 15 oocytes from each stage/group were collected.

### Library construction and sequencing

The SMART-Seq® HT kit (Takara Bio Inc., Shiga, Japan) was used for cDNA generation according to the manufacturer's instructions using 13 PCR cycles across a total of 15 samples. cDNAs were then converted to sequencing libraries using the Illumina DNA Prep kit (Illumina, San Diego, USA) according to the manufacturer's instructions with 8 PCR cycles for the final amplification. An equimolar pool of libraries was sequenced by single-end RNA sequencing (1 x 75bp) on an Illumina Nextseq 500 at the South Australian Genomics Centre (SAGC; Adelaide, SA).

### RNA-seq data collection and processing

Public human oocytes FASTQ sequencing files were downloaded from the Gene Expression Omnibus (GEO) database using the accession number GSE164371. All raw FASTQ files were removed adapters and low-quality bases using AfterQC tool [17], followed by quality control check by FastQC (version: 0.11.7). Only FASTQ files that can pass quality control were quantified at transcript level by *Kallisto* [16] in default settings. The reference transcriptome files (Sus_scrofa.Sscrofa11.1.cdna.all.fa and Homo_sapiens.genecode.V41.transcripts.fa) used for

building *Kallisto* index were downloaded from ENSEMBL and GENECODE database respectively.

For differential gene expression (DGE) analysis, we used R package tximport [18] to summarise *Kallisto* transcript-level estimates to gene level with the recommended scaling method lengthScaledTPM for limma-voom pipeline. Using gene-level counts, a DGEList object was created by DGEList function in R package edgeR [19], and filtered with filterByexpression function with specifying the group argument. The filtered DGEList object was transformed by voom function from R package limma [20], and then the voom object was processed with linear fit and Bayes correction to detect differentially expressed genes (DEGs). Differentially expressed genes were identified using the criteria $|\log_2(\text{fold change})| > 2$ and FDR < 0.01 in the pig dataset, and $|\log_2(\text{fold change})| > 1.5$ and FDR < 0.01 in the human dataset.

Additionally, we performed a differential transcript usage (DTU) analysis and a differential expression (DE) analysis based on transcript-level estimates [18]. DTU analysis was performed with R package IsoformSwitchAnalyzeR [21]. Transcriptome files used in the Kallisto alignment and corresponding gene transfer format (GTF) annotation were imported using importRdata function to establish a switchAnalyzeRlist object, followed by filtering out all genes and isoforms that do not suit the survival criteria (gene expression cut off = 1, IF cut off = 0.05, differential isoform fraction (dIF) cutoff = 0.1, removeSingleIsoformGenes = TRUE) by using preFilter function. The DTU events were tested by isoformSwitchTestDEXseq(reduceToSwitchingGenes = TRUE) function and significant DTU events were identified using the criteria FDR < 0.01 and $|\text{dIF}| > 0.1$. To predict alternative splicing (AS) and switching consequences, we performed external analyses such as CPC2 [22], Pfam [23], SingalIP [24] and IUPred2A [25] using the fasta files produced by extractSequence(removeLongAAseq = TRUE, alsoSplitFastaFile = TRUE, alpha = 0.01, dIFcutoff = 0.1, writeToFile = TRUE) function and then imported back to the switchAnalyzeRlist object using analyzeCPC2(codingCutoff = 0.725, removeNoncodinORFs = TRUE), analyzePFAM, analyzeSignalP and analyzeIUPred2A functions. To predict AS events and relevant switching consequence, analyzeAlternativeSplicing and analyzeSwitchConsequences were performed in default settings. Then, we used extractConsequenceSummary and extractSplicingSummary functions to quantify gain/loss (also shorter/longer, sensitive/insensitive, switch) of predicted functional consequences and gain/loss of splicing events. Also, to detect whether uneven usage of AS and switching consequence, extractSplicingEnrichment(countGenes = FALSE) and extractConsequenceEnrichment(countGenes = FALSE) were employed to do a binomial test and summary plot. Uneven usage was considered significant as FDR < 0.05.

## Gene ontology (GO) enrichment, canonical pathway and network analysis

The functional annotation for DEGs and genes with at least one significant DTU event (gDTUs) in the pig and human datasets were carried out by R package clusterProfiler [26]. For DEGs, GO enrichment analysis was done with up- or down-regulated genes respectively. The Benjamini–Hochberg algorithm was employed to control the false discovery rate (FDR) for multiple hypothesis. The GO pathways with FDR < 0.05 were considered significant.

Differentially expressed gene data sets determined by corrected *p* value (FDR<0.01) and $\log_2(\text{fold change}) > 2$ for the pig dataset and $> 1.5$ for the human dataset, and including up- and downregulated genes were imported into Ingenuity Pathway Analysis (IPA®, QIAGEN Redwood City, CA, United States). The DEGs were subjected to core analyses focussing on direct and indirect relationships pertaining to canonical pathways (metabolic and cell signalling), upstream regulators and network generation of differentially expressed gene interactions with other molecules within the Ingenuity Knowledge Base.

### Fisher exact statistical test

One tail Fisher exact statistical tests were performed by fisher.test function (alternative = "greater") in R to investigate whether there was significant overlapping ($P$ value $< 0.05$) of the DEGs on a specific pathway between pig dataset and human dataset. In the application of fisher.test function, the total number of genes on a specific pathway was the sum of the number of genes involved in the pathway (Gene Ontology database) of pig and human datasets.

### Data visualisation

The scatterplots and bar plots were generated with R package ggplot2. Heatmap and Venn diagram were drawn using R package pheatmap and VennDiagram, respectively. Exon-intron maps were generated using R package IsoformSwitchAnalyzeR.

### Data and code

Raw sequencing data, images, and processed data have been deposited in the Gene Expression Omnibus (GEO) repository under accession GEO: GSE237026. All original code has been deposited at GitHub: https://github.com/tf1993614/Pig-Human-RNAseq-analysis/tree/main.

## Results

### Transcriptome characterisation of pig and human oocytes

To investigate the gene expression changes across oocyte maturation, we performed RNAseq analysis on isolated and *in vitro* matured pig oocytes and re-analysed public RNAseq data from human oocytes of different development stages for comparison.

Quality control of our pig datasets identified a GV stage oocyte sample, which only comprised 13 million sequencing reads compared to on average 20 million sequencing reads in the other samples. This GV sample was removed from downstream analyses. We detected 22,516 in the pig dataset and 62,610 genes in the human dataset due to the different sequencing platforms used. After filtering out genes with low expression ($< 10$ counts per million (pm)), 12,962 and 18,998 genes were preserved in pig and human dataset, respectively, for further application.

Principle component analysis (PCA) showed that GV oocytes were significantly different from MII oocytes in both datasets (Fig 1A and 1C). Surprisingly, in the pig dataset, MII and 'damaged' oocytes did not separate in the PCA (Fig 1A). Therefore, we combined these two groups for further analyses ('MIX'). Consistent with PCA, the distance matrix (Pearson correlation) across samples also revealed similar relationships between oocytes of a distinct development phase in both species (S2A and S2C Fig). A significant different gene expression pattern between GV oocytes and MII oocytes was also observed in the heatmaps of both species (S2B and S2D Fig).

### The effect of developmental stage on the oocyte gene expression

Differential gene expression analysis comparing MIX (MII + 'damaged') versus GV stage oocytes in the pig dataset found 2,654 differentially expressed genes (DEGs), of which 538 genes were upregulated and 2,116 genes were downregulated in MIX (MII + 'damaged') oocytes as shown in the Volcano plot (Fig 1B). In the human dataset, we found 3,969 DEGs, of which 1,118 genes were upregulated and 2,851 downregulated in MII oocytes compared with GV oocytes (Fig 1D). The complete list of DEGs for both species are presented in S1 and S9 Tables. We did not find any DEGs comparing 'MII' with 'damaged' in the pig dataset (S3A Fig) strengthening our decision to combine these two groups.

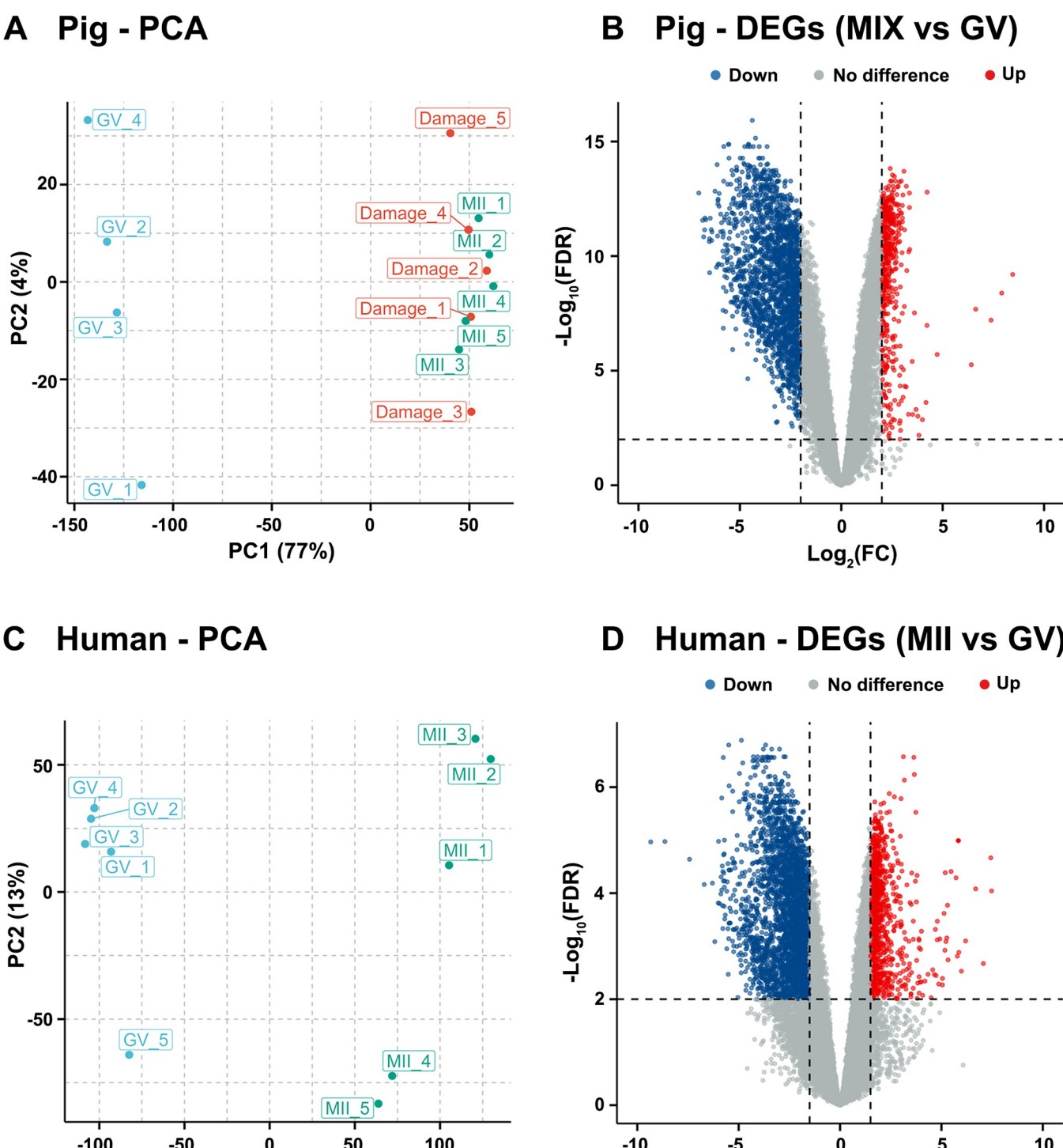

**Fig 1. Principal Component Analysis (PCA) and analysis of differentially expressed genes (DEGs) of oocyte samples across different developmental stages in the pig and human datasets.** (A) PCA analysis of pig oocytes showing GV oocytes (n = 4) in blue, while 'damaged' (n = 5) and MII oocytes (n = 5) are in red and green, respectively. (B) Volcano plot showing the differentially up- and downregulated genes (FDR < 0.01 and |log$_2$(fold change)| > 2) between pig 'Mix' (MII and 'damaged' oocytes combined) and GV stage oocytes. (C) PCA analysis of human oocytes showing GV oocytes (n = 5) in blue and MII oocytes (n = 5) in green. (D) Volcano plot showing the differentially up- and downregulated genes (FDR < 0.01 and |log$_2$(fold change)| > 1.5) between human MII and GV stage oocytes. Upregulated and downregulated genes are coloured in red and blue, respectively.

We found 53 DEGs to be associated with the oocyte and the cell cycle in the comparison of MIX (MII + 'damaged') versus GV oocytes in the pig, whereat 36 were up- and 17 downregulated (S2 Table). In the human comparison, 89 genes related to oocyte and cell cycle were differentially expressed, whereat 51 were up- and 38 downregulated (S10 Table). Surprisingly, the transcription factor *POU5F1*, which is playing a key role in stem cell pluripotency and embryonic development, is upregulated, whereas another molecule associated with embryonic development, *NODAL*, is downregulated (S10 Table). DEGs associated with mitochondria and oxidative phosphorylation were mostly downregulated (97 genes) in the pig comparison except for coenzymes *COQ10A* and *COQ10B*, mitochondrial ribosomal protein L44 (*MRPL44*) and the mitochondrial NUBP iron-sulfur cluster assembly factor (*NUBPL*) (S3 Table). In the human comparison, mitochondrial-associated genes were more evenly up- (38 genes) and downregulated (59 genes) (S11 Table). Steroidogenesis related genes are mostly unaffected by oocyte maturation in the pig (3 genes up- and 13 genes downregulated) or human (5 genes up- and 5 genes downregulated) (S4 and S12 Tables, respectively). The transition of GV oocytes into MII oocytes is connected with an upregulation of catalase (*CAT*) and glutathione peroxidase 1 (*GPX1*), whereas other antioxidant enzymes, such as peroxiredoxins *PRDX1*, *PRDX3*, and *PRDX5*, nucleoredoxin (*NXN*), thioredoxin 2 (*TXN2*), *GPX4*, *GPX6*, and glutaredoxin 3 (*GLRX3*) are downregulated in the pig comparison (S5 Table). In the human, however, *PRDX3* and *PRDX4* are upregulated, whereas thioredoxin reductase 2 (*TXNRD2*), *GLRX*, *GLRX3* and *NXN* are downregulated (S13 Table). We found some DEGs to be associated with endoplasmic reticulum (ER) stress. The majority of these genes was downregulated in MIX (MII + 'damaged') oocytes in the pig (S6 Table) and in the human (S14 Table). All DEGs related to ribosomal structure, such as ribosomal proteins for the small and large subunit, were downregulated in MIX (MII + 'damaged') versus GV oocytes in the pig (S7 Table), whereas in human 50% were up- and 50% downregulated (S15 Table). Overall, 40 genes in the pig MIX (MII + 'damaged') oocytes and 53 in the human MII oocytes are differentially expressed and associated with the cytoskeleton and extracellular matrix compared with GV oocytes (S8 and S16 Tables respectively). Upregulated DEGs include collagen type XIX alpha 1 (*COL19A1*) and versican (*VCAN*) in the pig (S8 Table), and *COL1A1*, fibronectin 1 (*FN1*), laminin subunits alpha 2 (*LAMA2*) and beta 1 (*LAMB1*) in the human comparison (S16 Table). Downregulated genes include, amongst others, *COL28A1*, nidogen 1 (*NID1*), *COL15A1*, *COL23A1*, and brevican (*BCAN*) in the pig (S8 Table), and *COL4A6*, *COL18A1* as well as *COL23A1* in the human (S16 Table).

To determine which biological functions are connected with these DEGs, gene ontology (GO) enrichment analysis was carried out using upregulated or downregulated DEGs separately in both datasets (Fig 2). In the pig dataset, as expected, upregulated DEGs were associated with pathways related to cell division (meiosis) including chromosome structure and segregation, cell cycle, spindle formation and DNA repair mechanisms (Fig 2A), while downregulated DEGs were involved with organelle structures and biogenesis such as mitochondria, endoplasmic reticulum and ribosomes structural molecule activity (Fig 2B). Consistent with the pig dataset, the human dataset, revealed upregulated DEGs to be enriched in pathways including chromosome structure and segregation, histone binding and nucleosome organisation (Fig 2C). However, in human also ribosomal organisation and processes were enriched by upregulated DEGs. This is contrary to the pig. Interestingly, in the human dataset, only one significant pathway, mitochondrial matrix, was enriched for downregulated DEGs (Fig 2D). To identify if there is significant overlap between the GO pathways enriched for upregulated/downregulated DEGs between the two species, we performed a Fisher exact test. We found only DEGs involved in the "chromosome segregation" pathway showed a high correlation ($-\log_{10} p$ value > 1.3) (Fig 3).

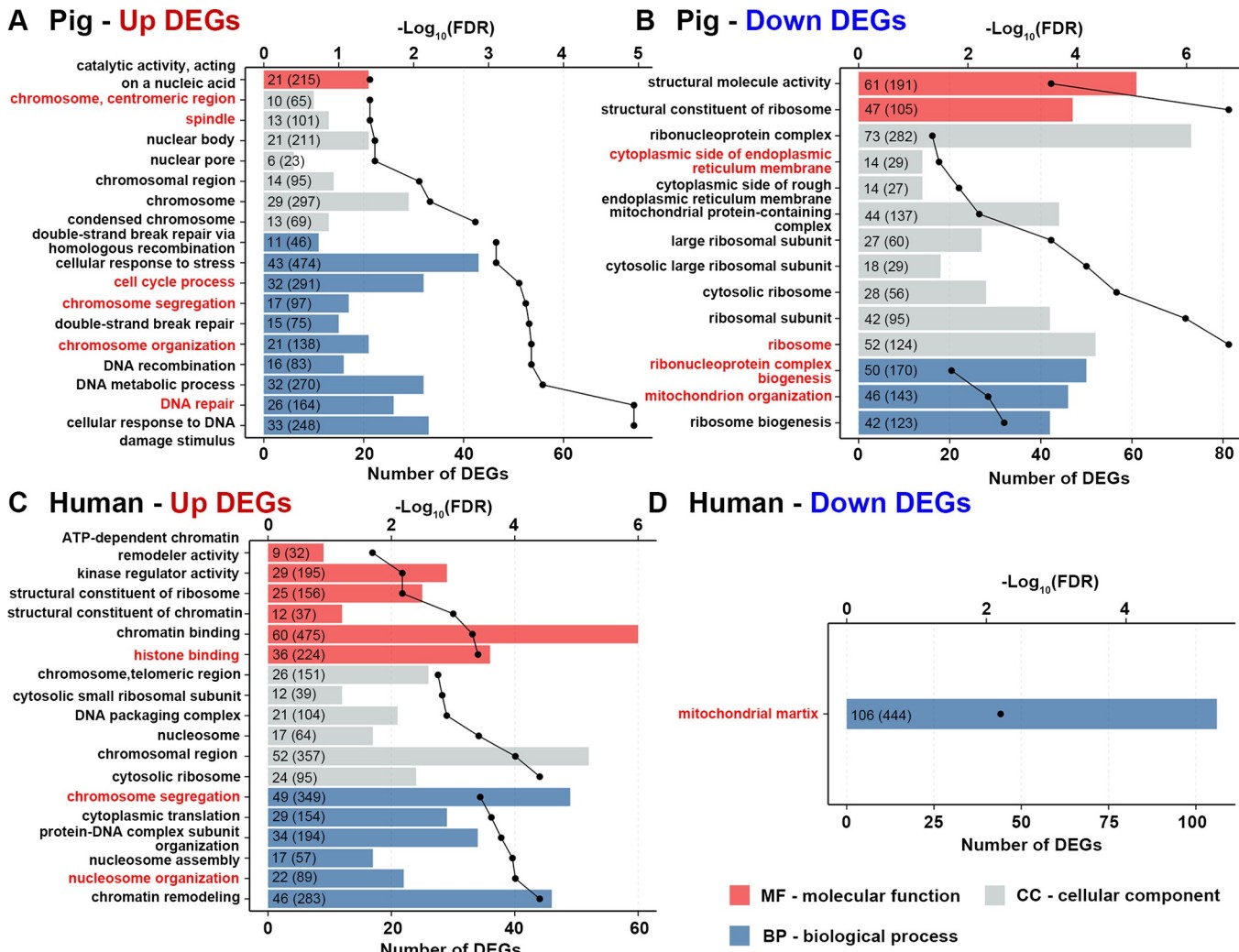

**Fig 2. GO enrichment analysis of differentially up- and downregulated genes in the pig and human datasets.** Significant GO pathways enriched in differentially upregulated genes (A) and downregulated genes (B) by comparing pig MIX (MII + 'damaged') versus GV oocytes. Significant GO pathways enriched in differentially upregulated genes (C) and downregulated genes (D) by comparing human MII versus GV oocytes. The dotted line represents the FDR of each significant GO pathway. The numbers in the bar outside the parentheses represent the number of DEGs involved in the pathway, whereas inside the parentheses indicates the total number of genes associated with the pathway in the GO database. Only pathways with FDR < 0.05 were considered significant. Bars are coloured by three GO categories: molecular function (MF) in red, cellular component (CC) in grey and biological process (BP) in blue.

Furthermore, IPA analysis of up- and downregulated genes combined showed relevant and similar canonical pathways (Fig 4) as found by GO enrichment analysis (Figs 2 and 3). The top canonical pathways associated with mitochondria dysfunction, chromosome segregation, signalling pathways associated with metabolism such as sirtuin signalling, mTOR signalling and NAD signalling, nucleotide excision repair and oxidative phosphorylation were involved in DEGs identified in both datasets (Fig 4).

However, lipid metabolism-relevant pathways, including different cholesterol biosynthesis pathways and geranylgeranyldiphosphate biosynthesis, only appeared in the pig dataset (Fig 4A). In addition, top upstream regulators associated with DEGs in both datasets were identified using IPA and are shown in Tables 1 and 2, respectively. Biological upstream regulators such as HNF4A, COPS5, TP53 and MAP4K4 were common to both datasets. Chemical upstream regulators included Torin 1, 3,5-dihydroxyphenylglycine, Acyline, LH, Metribolone,

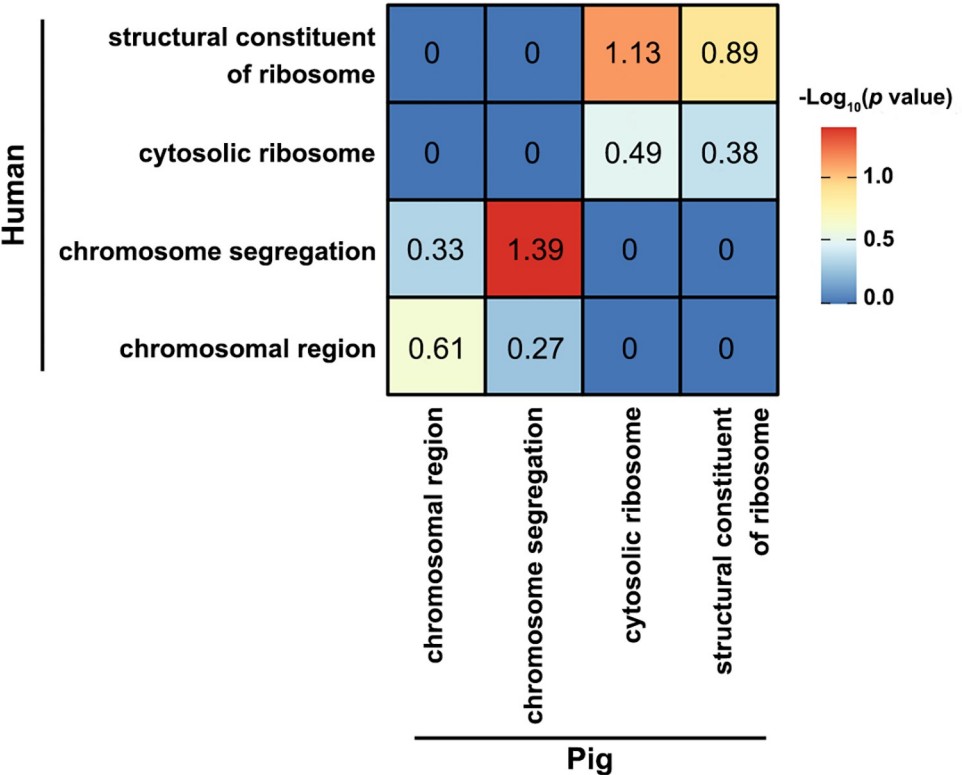

**Fig 3. Correlation of differentially expressed genes (DEGs) involved in identical significant GO pathways in the pig and human datasets.** Fisher exact tests were conducted to determine a significant overlap of DEGs in a specific pathway between the two datasets. The number and gradient colour of the legend bars represent $-\log_{10} P$ value of each pairwise comparison. Only $P$ value $< 0.05$ was considered significant.

ST1926 and sirolismus in the pig dataset, and Actinonin, CD437, GSKJ4 and ST1926 in the human dataset. Interestingly, the upstream regulators for the human include numerous mitochondria-related molecules such as mitochondrial ribosomal protein L12 (MRPL12) and L14 (MRPL14), enzymes lon peptidase 1 (LONP1), NOP2/Sun RNA methyltransferase 3 (NSUN3) and ClpB family mitochondrial disaggregase (CLBP), GTP dependent ribosome recycling factor mitochondrial 2 (GFM2), and mitochondrially-encoded t-RNAs (MT-TM, MT-TE). Full upstream regulator lists for both species are shown in S17 and S20 Tables.

Additionally, network analysis was performed in IPA for DEGs in each dataset showing their function. Networks of interest are presented in Fig 5 for the pig dataset including cell cycle progression (network 3), spindle assembly (network 6), and mitochondrial ribosomal proteins involved in the assembly and stability of nascent mitochondrial polypeptides exiting the ribosome (network 7). Network 1 contains retinoid acid-inducible gene 1 (RIG-I), as central molecule, which is a pattern recognition receptor for the innate immune response.

Important networks from the human dataset are shown in Fig 6. Mitochondrial processes such as β-fatty acid oxidation pathway (e.g. LACTB/ lactamase B, ACAD9/ acyl-CoA dehydrogenase family member 9, ECHS1/ enoyl-CoA hydratase, short chain 1), oxidative phosphorylation (e.g. ACAD9, ATPAF2/ ATP synthase mitochondrial F1 complex assembly factor 2), and response to oxidative stress (e.g. PYCR1, PRDX3). Caseinolytic mitochondrial matrix peptidase proteolytic subunit (CLPP) is involved in ATP-dependent protein/peptide cleavage via Clp complex and cyclin dependent kinase 2 associated protein 2 (CDK2AP2) is suggested to regulate microtubule organisation in MII oocytes. Networks 6 and 18 contain molecules

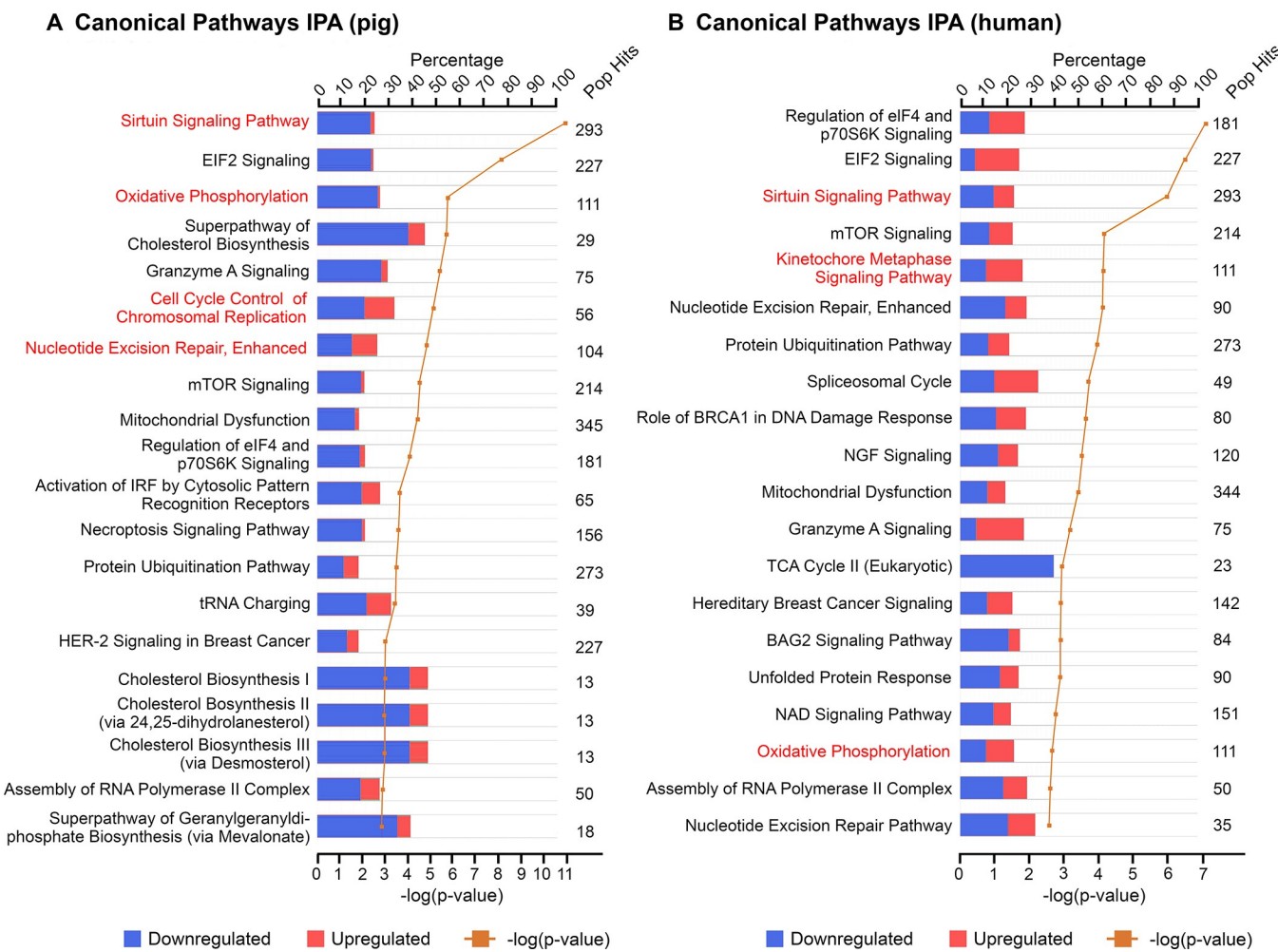

**Fig 4. Top 20 canonical pathways associated with differentially expressed genes (DEGs) in the pig and human datasets as identified using IPA.** (A) Top 20 canonical pathways involved in DEGs in pig dataset. (B) Top 20 canonical pathways involved in DEGs in human dataset. "Pop Hits" refers to the total number of genes associated with each of the pathways in the database. The bar graphs represent the percentage of genes from the data set that map to each canonical pathway whilst the orange line shows the *P* value of overlap between DEGs in each dataset and a given pathway.

involved in ribosome function, including rRNA processing (e.g. ribosomal proteins RPL8, RPL28, and RPL10, RPL27, respectively), RNA modification (e.g. MEPCE/ methylphosphate capping enzyme), t-RNA splicing (e.g. RTCB/ RNA 2',3'-cyclic phosphate and 5'-OH ligase), and pre-mRNA splicing (e.g. SRSF6/ serine and arginine rich splicing factor 6; RNF113A/ ring finger protein 113A), as well as chromatin remodelling and transcription (e.g. HMGB2/ high mobility group box 2), and centrosome duplication (e.g. NPM1/ nucleophosmin 1). Network 25 includes molecules involved in DNA damage response and maintenance of normal chromosome stability such as members of the Fanconi anemia complementation group and centromere proteins. Detailed information of the genes involved in the networks can be found in S18, S19, S21 and S22 Tables.

## The effect of developmental stage on the oocyte transcript isoform usage

Classic DGE analysis fails to detect genes, which show differences in isoform transcript levels but result in no difference at gene level between two conditions. Therefore, we used differential transcript usage (DTU; also called isoform switching) analysis to identify differences between

**Table 1. Top 20 upstream regulators in the pig dataset.** Shown are respective activation z-score as well as *P* value of association for the DEGs.

| Upstream regulator | Name | z-score | *p*-value |
|---|---|---|---|
| Torin 1/ mTOR inhibitor XI | 1-(4-(4-propionylpiperazin-1-yl)-3-(trifluoromethyl)phenyl)-9-(quinolin-3-yl)benzo[h][1,6] naphthyridin-2(1H)-one | 4.683 | 3.44E-19 |
| LARP1 | La ribonucleoprotein 1 | 6.243 | 6.11E-18 |
| HNF4A | Hepatocyte nuclear factor 4 alpha | -3.838 | 7.95E-18 |
| MLXIPL | MLX interacting protein like | -6.202 | 2.52E-12 |
| COPS5 | COP9 signalosome subunit 5 | 5.077 | 9.30E-12 |
| RICTOR | RPTOR independent companion of MTOR complex 2 | 6.647 | 8.54E-11 |
| MYCN | MYCN proto-oncogene, bHLH transcription factor | -2.92 | 1.61E-10 |
| TP53 | Tumor protein p53 | -0.129 | 1.29E-09 |
| 3,5-dihydroxyphenylglycine | - | -5.064 | 2.41E-09 |
| KDM5A | Lysine demethylase 5A | 6.083 | 3.39E-09 |
| Acyline | - | - | 5.53E-09 |
| MYCL | MYCL proto-oncogene, bHLH transcription factor | -1.673 | 6.07E-09 |
| LH | Luteinizing hormone | -4.234 | 6.3E-09 |
| TLE3 | TLE family member 3 | 1 | 1.28E-08 |
| Metribolone/ Androgen receptor agonist | 17alpha-Methyltrienolone | -4.281 | 1.92E-08 |
| ST1926/ Adarotene/ Apoptosis inducer | (2E)-3-(4′-hydroxy-3′-tricyclo[3.3.1.13,7]dec-1-yl[1,1′-biphenyl]-4-yl)-2-propenoic acid | 5.690 | 3.27E-08 |
| Sirolismus | - | 3.405 | 5.22E-08 |
| MAP4K4 | Mitogen-activated protein kinase kinase kinase kinase 4 | 4.596 | 7.35E-08 |
| UQCC3 | Ubiquinol-cytochrome c reductase complex assembly factor 3 | - | 1.12E-07 |
| RB1 | RB transcriptional corepressor 1 | -4.139 | 2.1E-07 |

oocytes at different developmental stages at isoform transcript level instead of whole gene level. In the pig dataset, 11,538 transcript isoforms were kept for DTU analysis. Comparing 'MIX' (MII + 'damaged') oocytes with GV oocytes identified 2,951 DTU events involving 4,494 transcript isoforms of which 2,256 isoforms had upregulated usage and 2,238 isoforms had downregulated usage in MIX (MII + 'damaged') and 2,217 genes (those genes with DTU events referred to gDTUs) (Fig 7A and 7B). All 2,217 gDTUs were protein-coding and only 270 gDTUs overlapped with DEGs found by traditional DGE analysis (Fig 7B). GO enrichment analysis of 1,947 gDTUs, only identified by DTU analysis, revealed six significant pathways mainly involved in intracellular transport mechanisms (Fig 7C). Interestingly, DTU analysis also identified two genes (*RAB4B* and *DNTBP1*), which had significantly decreased isoform usage when comparing MII with 'damaged' (S3B–S3D Fig). These DTU events took place in the longest transcript isoform of both genes (S3E and S3F Fig).

**Table 2. Top 20 upstream regulators in the human dataset.** Shown are respective activation z-score as well as *P* value of association for the DEGs.

| Upstream regulator | Name | z-score | *p*-value |
|---|---|---|---|
| HNF4A | Hepatocyte nuclear factor 4 alpha | -2.291 | 2.19E-13 |
| DAP3 | Death associated protein 3 | 3.606 | 9.32E-13 |
| LONP1 | Lon peptidase 1, mitochondrial | 2.774 | 5.14E-12 |
| COPS5 | COP9 signalosome subunit 5 | 4.707 | 1.44E-11 |
| Actinonin/ antibacterial agent | - | -3.742 | 5.29E-11 |
| MT-TM | Mitochondrially encoded tRNA methionine | - | 7.75E-11 |
| TP53 | Tumor protein p53 | 1.168 | 2.67E-09 |
| ALKBH7 | alkB homolog 7 | 3.130 | 2.94E-08 |
| NSUN3 | NOP2/Sun RNA methyltransferase 3 | 2.828 | 3.97E-08 |
| MT-TE | Mitochondrially encoded tRNA glutamic acid | - | 3.35E-07 |
| CD437/ RARγ-selective agonist | 6-(4-Hydroxy-3-tricyclo[3.3.1.13,7]dec-1-ylphenyl)-2-naphthalenecarboxylic acid | -0.608 | 4.83E-07 |
| GSKJ4/ histone lysine demethylase inhibitor III | Ethyl-3-(6-(4,5-dihydro-1H-benzo[d]azepin-3(2H)-yl)-2-(pyridin-2-yl)pyrimidin-4-ylamino)propanoate | -0.392 | 1.65E-06 |
| ST1926/ Adarotene/ apoptosis inducer | (2E)-3-(4′-hydroxy-3′-tricyclo[3.3.1.13,7]dec-1-yl[1,1′-biphenyl]-4-yl)-2-propenoic acid | -0.206 | 1.83E-06 |
| MRPL12 | Mitochondrial ribosomal protein L12 | 2.613 | 2.40E-06 |
| MAP4K4 | Mitogen-activated protein kinase kinase kinase kinase 4 | 4.490 | 1.42E-05 |
| MRPL14 | Mitochondrial ribosomal protein L14 | 1.633 | 1.77E-05 |
| HMGXB4 | HMG-box containing 4 | -1.038 | 2.27E-05 |
| GFM2 | GTP dependent ribosome recycling factor mitochondrial 2 | 2.236 | 2.37E-05 |
| CLPB | ClpB family mitochondrial disaggregase | 2.309 | 5.77E-05 |
| miR-2392 (miRNAs w/seed AGGAUGG) | - | -2.441 | 6.37E-05 |

In the human dataset, 55,459 transcript isoforms remained for DTU analysis after quality control. Comparison of MII oocytes with GV oocytes resulted in 3,932 DTU events involving 4,672 transcript isoforms of which 2,249 isoforms had upregulated usage and 2,423 isoforms downregulated usage in MII, and 2,623 gDTUs (Fig 8A and 8B). Similarly, only 549 gDTUs overlapped with DEGs (Fig 8B). Unlike the pig dataset, in the human dataset only 88% gDTUs were protein-coding, whereas the other 12% were assigned to long non-coding RNAs (lncRNAs) and transcribed unprocessed/processed/unitary pseudogenes (Fig 8C). Significant pathways such as GTPase binding, ubiquitin protein ligase binding, regulation of protein stability and mRNA processing were enriched in the 2,074 gDTUs only identified by DTU analysis (Fig 8D).

To identify the underlying causes of the transcript isoform switches, we quantified the number of splicing events associated with isoform switches found in both species datasets. In the pig dataset, we found that alternative transcription start site (ATSS), alternative transcription termination site (ATTS), and exon skipping (ES) were the most frequent splicing events (S4A and S4B Fig) and ATTS was the only splicing event enriched in upregulated isoform switches (S4C Fig). Further investigation also revealed that many isoforms lost coding potential and protein domains (S4D and S4E Fig). A similar phenomenon was observed in the human dataset (S5 Fig).

## Discussion

Exploring gene expression changes during oocyte maturation is essential to understand their molecular role in the regulation of the transition of GV oocytes to MII oocytes and the associated completion of meiosis I. In this study, we established a high-resolution transcriptome

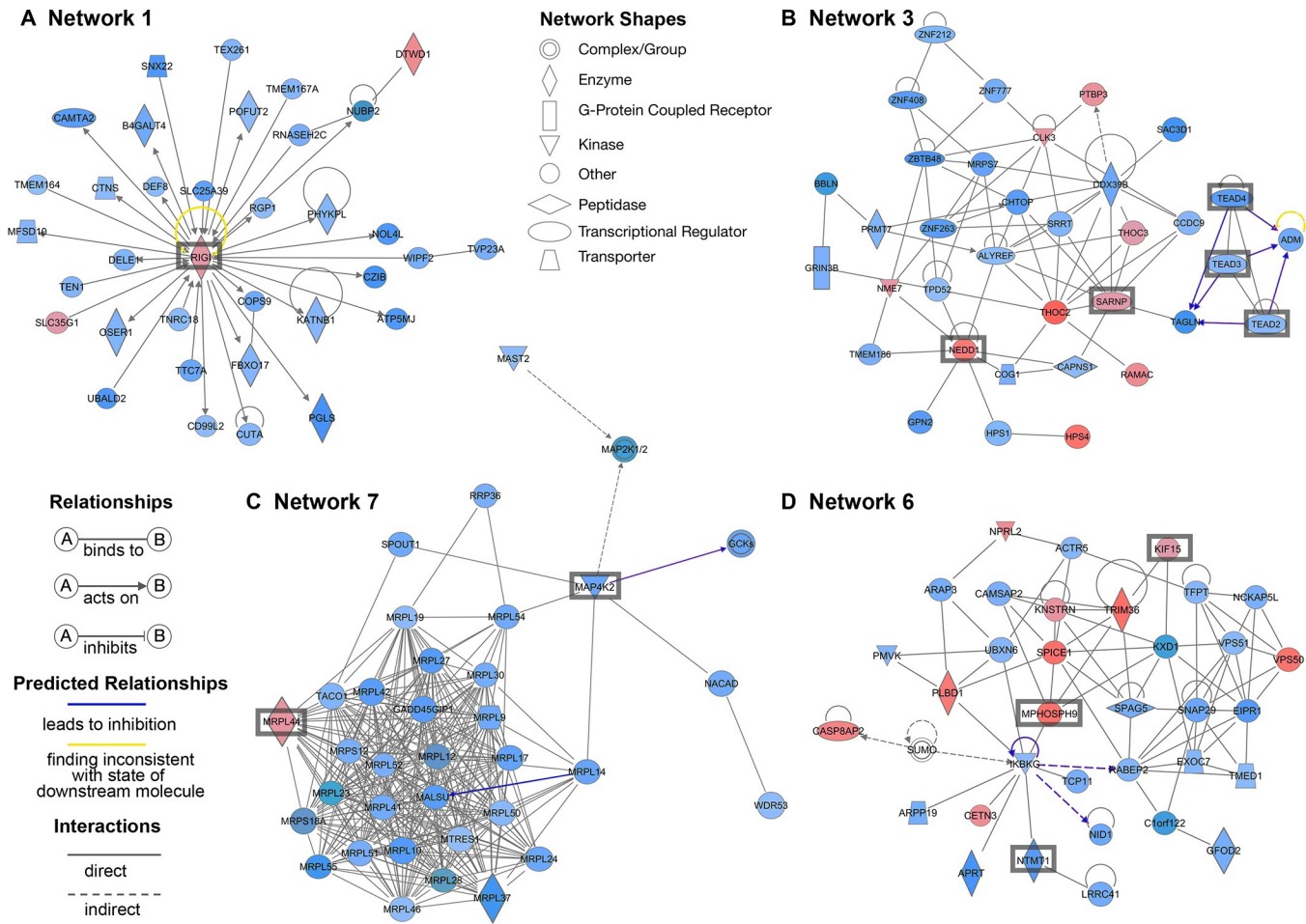

**Fig 5. Networks associated with differentially expressed genes (DEGs) in the pig dataset as identified by IPA.** Network 1 (A), network 3 (B), network 6 (C) and network 7 (D) associate with immune response, cell cycle progression, mitochondrial ribosomal proteins and spindle assembly, respectively. Blue colour represents low expression; red colour represents high expression in MIX (MII + 'damaged') oocytes. The intensity of each colour shows the strength of regulation.

profile of the pig oocyte at three developmental stages (GV, MII and 'damaged') and of the human oocyte at two stages (GV and MII), identifying key regulators, which drive oocyte maturation in both species. In the pig dataset since the MII with a polar body and the 'damaged' oocytes without a polar body did not separate in the PCA we combined these two groups for further analyses and referred to them as 'MIX'.

In mammals, oocytes undergo a unique asymmetric cell division of meiosis to facilitate their maturation. This process involves the resumption of a long-term dormant stage referred to as GV, followed by gradual progression towards MII after accepting activation signals. As part of the first meiosis, the oocyte extrudes its first polar body to complete the process [27]. MII oocytes stay arrested at metaphase II until fertilisation occurs [27]. To get healthy MII oocytes with fertilisation competence, involves a series of biological events such as spindle assembly, chromosome alignment and chromosome segregation during meiosis I. It has been reported that disruption of meiotic spindle assembly prevents oocytes from transitioning from GV to MII in various species [3, 4, 28], indicating that an intact spindle structure is essential for oocyte maturation. The kinesin family, a class of ATP-dependent microtubule-based motor proteins, plays an important role in maintaining a stable spindle. Depletion of *Kif4a*,

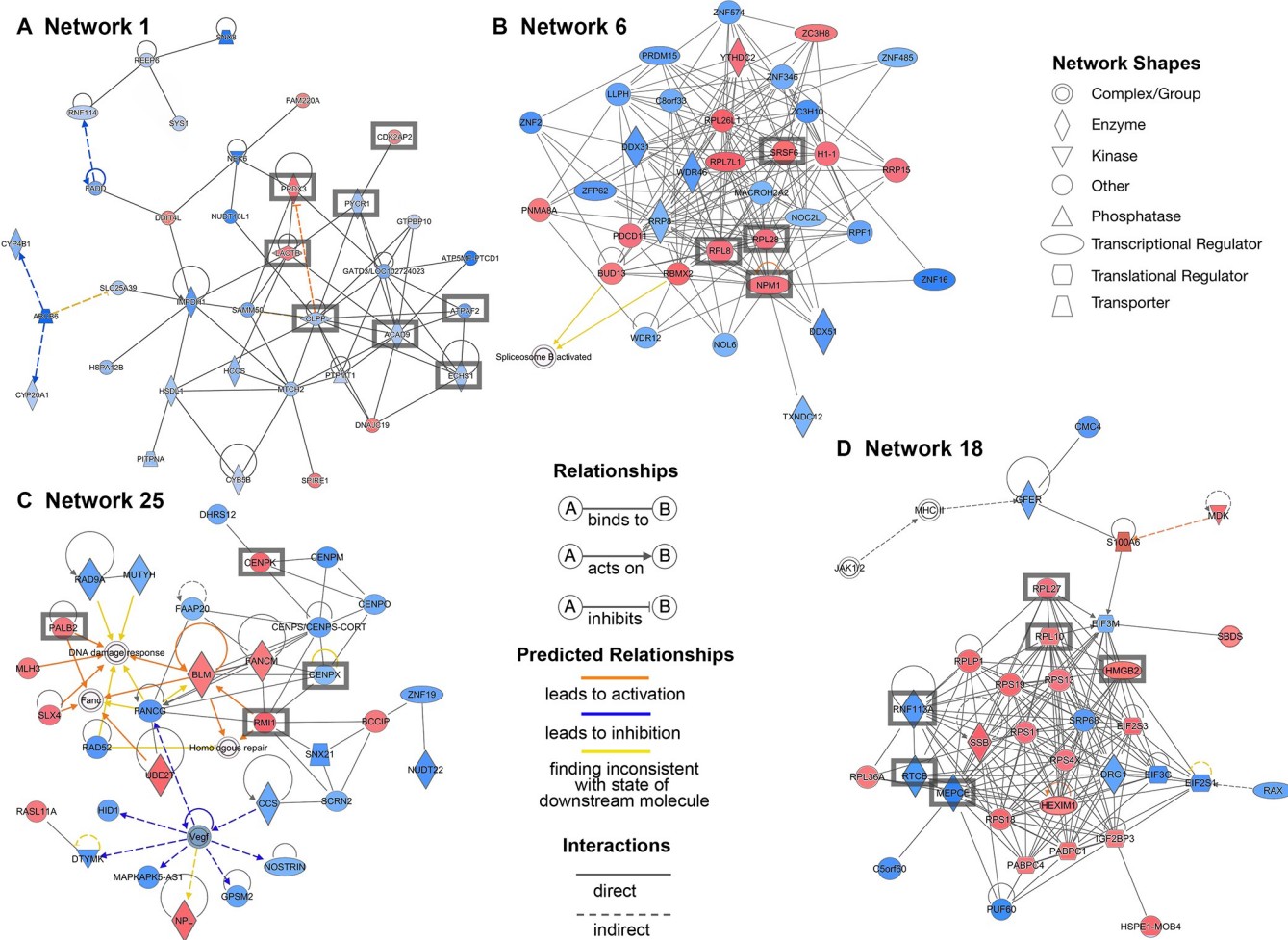

**Fig 6. Networks associated with differentially expressed genes (DEGs) in the human dataset as identified in IPA.** Network 1 (A), network 6 (B), network 25 (C) and network 18 (D) associated with mitochondrial processes, ribosome function, DNA damage response and maintenance of normal chromosome stability, respectively. Blue colour represents low expression; red colour represents high expression in MII oocytes. The intensity of each colour shows the strength of regulation.

*Kif18a* in mouse [3, 29] or *KIF11* in pig [4] disrupts spindle formation and then leads to oocyte maturation arrest. Besides, inhibition of *Kif15* diminishes spindle microtubule stability in mouse oocytes [30]. Consistent with this, we found in this study that kinesin genes such as *KIF15*, *KIF18A* and *KIF20B* in pig MIX oocytes and *KIF4B*, *KIF5B*, *KIF5C* as well as *KIF20B* in human MII oocytes had significantly higher expression, suggesting that those kinesins were highly expressed during GV to MII transition, thereby facilitating oocyte maturation. Additionally, other genes including *NEDD1* and *CEP192*, which have been shown to be required for spindle assembly in the mouse oocyte [31, 32], were expressed higher in pig MIX and human MII oocytes, respectively, compared to the GV stage. Furthermore, *SGO1*, *SGO2*, *CDC40* or *CDC42*, key genes regulating chromosome segregation and cell cycle as previously reported [33–36], also had higher expression in MIX/MII oocytes in our pig dataset and in MII in the human dataset. Genes involved in chromosome segregation appear to be conserved during oocyte maturation between pig and human.

Unlike somatic cells where transcription persists in mitosis [37], oocytes cease transcription once they resume meiosis and continue progression from GV stage onwards. This means the

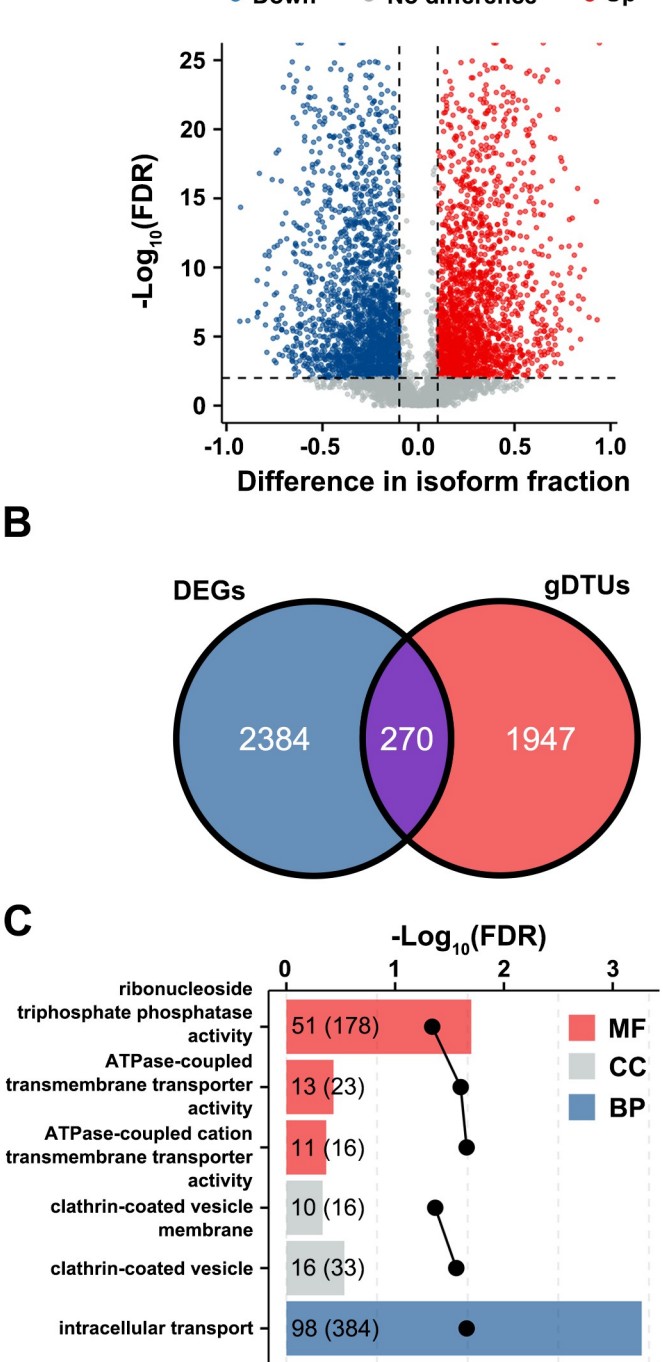

**Fig 7. DTU analysis for comparison of pig 'MIX' (MII + 'damaged') oocytes with GV oocytes.** (A) Differentially expressed isoform transcripts involved in isoform switching. Isoforms with upregulated isoform usage (differential isoform fraction (dIF) > 0.1 and FDR < 0.01) are shown in red, isoforms with downregulated isoform usage (dIF < -0.1 and FDR < 0.01) in blue. (B) Venn diagram showing the number of genes common to gene lists identified by DGE and DTU analysis. (C) Top 6 differential GO pathways enriched in the gDTUs as identified by DTU analysis. The dotted line represents the FDR of each significant GO pathway. The numbers in the bars outside the parentheses

represent the number of gDTUs involved in the pathway and those in parentheses indicate the total number of genes enriched for the pathway in the dataset. Only pathways with FDR < 0.05 were considered significant. Bars are coloured red for molecular function (MF), grey for cellular component (CC) or blue for biological process (BP).

oocyte can only recruit dormant maternal mRNAs for translation during maturation [38, 39]. In other words, prior to meiotic resumption oocytes must synthesise and store large numbers of mRNAs and ribosomes for later use until embryonic genome activation (EGA) [40]. Previous microarray analysis in mouse found that ribosomal gene expression is down-regulated in MII oocytes compared to GV oocytes and most of them are prone to degradation during GV to MII transition [41]. In line with that observation, we also found that all 79 differentially expressed ribosomal genes were downregulated in pig MIX oocytes. This may imply that GV oocytes have more potential for message translation compared to MII oocytes.

In the human dataset, only about 50% of differentially expressed ribosomal genes were downregulated, suggesting that MII oocytes may retain translation potential in human. Our further investigation also showed that DEGs involved in ribosome-relevant pathways in pig oocytes were not the same as in human oocytes. The time point of EGA might explain this different regulation of ribosomal genes across oocyte maturation between different species. It occurs at the two-cell stage in mouse [42], at the four-cell stage in pig [43, 44] and in human at the latest, the eight-cell stage [45]. This indicates that the time relying on maternal mRNA storage to support early embryo development would be longer in human than in pig and mouse. Therefore, one hypothesis is that human MII oocytes need to maintain a relatively high expression of ribosomal genes for the potential mRNA translation during early embryo development after fertilisation. In fact, translation of maternal mRNAs throughout oocyte maturation, fertilisation and embryogenesis is complex [46] and the underlying mechanism is still elusive.

Oocyte maturation is an energy-intensive process, therefore accurate regulation of mitochondrial function and the antioxidant system is critical [47]. There is a positive relationship between oocyte developmental competence and the amount of mitochondrial DNA (mtDNA) and ATP [7]. From GV to MII transition in human oocytes, mtDNA copies increase from 10,000 to approximately 400,000 per oocyte [48] and simultaneously the average ATP concentration reaches 2 pmol per oocyte [7], which is consistent with the previous findings that the human early GV oocyte has low basal respiration rates [49]. Mitochondrial dysfunction and reduction of ATP content lead to aneuploidy and spindle disassembly as well as chromosome misalignment in MII oocytes [8, 9]. Therefore, keeping mitochondria-based ATP production normal is the key to oocyte maturation.

Mitochondria rely on their electron transport chain (ETC) consisting of four complexes to synthesise ATP via oxidative phosphorylation (OXPHOS) [50]. These complexes are encoded by 92 genes, of which 79 genes are encoded by nuclear DNA and 13 genes by mtDNA [51]. Although nuclear DNA makes a significant contribution due to encoding more ETC components, mutation of mtDNA-encoded genes during oogenesis also results in cellular, tissue and organ dysfunction in the resulting offspring [52]. Therefore, understanding how nuclear DNA- and mtDNA-encoded OXPHOS genes synchronously determine mitochondrial function and the dynamics of gene expression during oocyte maturation is meaningful. In this study, our analysis showed that most of mtDNA-encoded and nuclear DNA-encoded OXPHOS genes were upregulated and downregulated, respectively, from GV to MII transition in both species. The downregulation of most nuclear DNA-encoded OXPHOS genes may be associated with decreasing mitochondrial workload in MII oocytes compared to GV oocytes, which has been proposed in bovine oocytes [53]. To keep homeostasis by counteracting ROS due to electron leakage from mitochondrial ETC, oocytes must express antioxidant genes

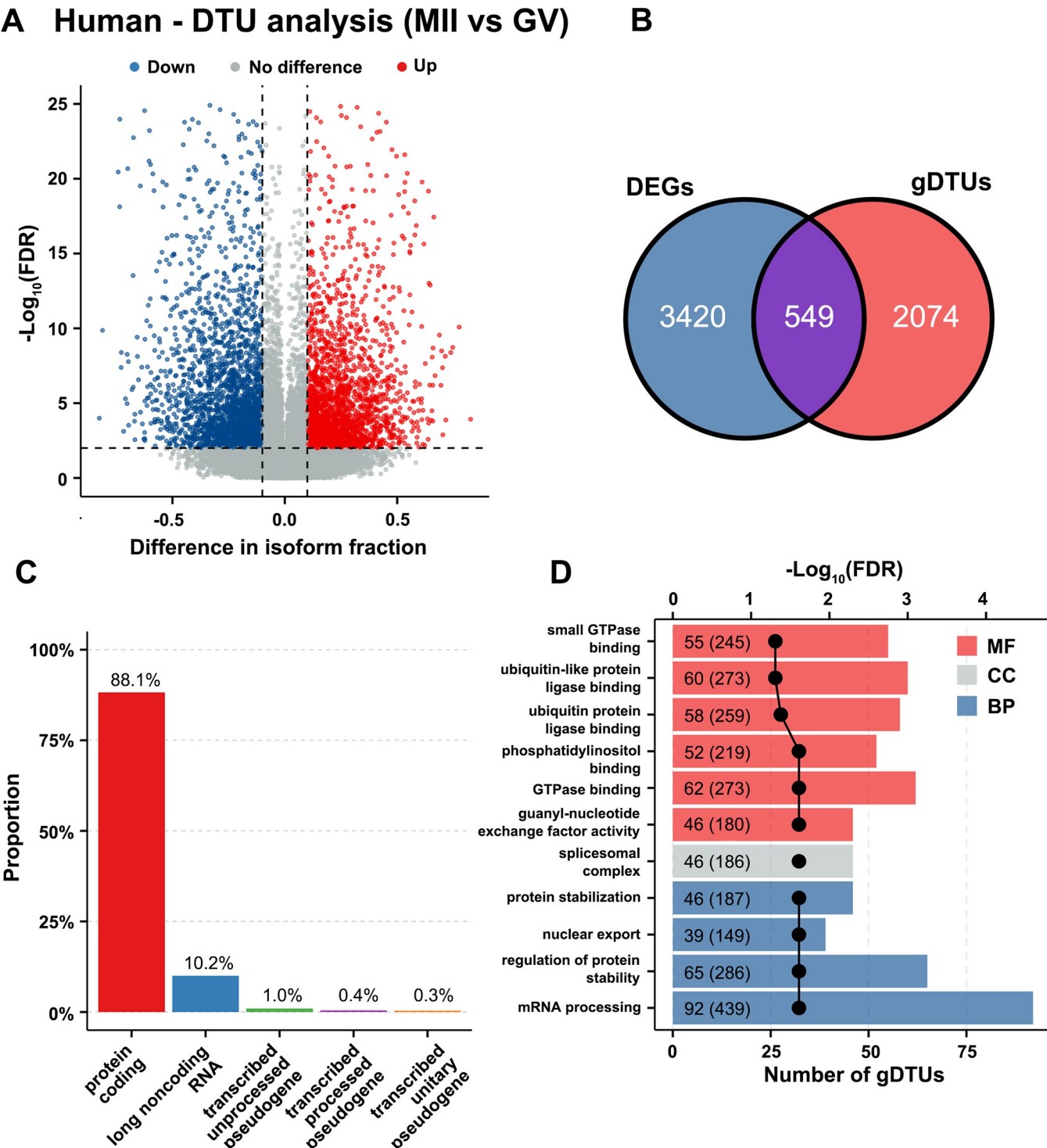

**Fig 8. DTU analysis for comparison of human MII oocytes with GV oocytes.** (A) Differential transcript isoforms involved in isoform switching. Isoforms with upregulated isoform usage (differential isoform fraction (dIF) > 0.1 and FDR < 0.01) are shown in red, isoforms with downregulated isoform usage (dIF < -0.1 and FDR < 0.01) in blue. (B) Venn diagram showing the number of genes common to gene lists identified by DGE and DTU analysis. (C) Proportion of biotypes of gDTUs found by DTU analysis. (D) Top 13 significant GO pathways enriched in the gDTUs only identified by DTU analysis. The dotted line represents the FDR of each significant GO pathway. The numbers in the bars outside the parentheses represent the number of gDTUs involved in the pathway and those in parentheses indicate the total number of genes enriched for the pathway in the dataset. Only pathways with FDR < 0.05 were considered significant. Bars are coloured red for molecular function (MF), grey for cellular component (CC) or blue for biological process (BP).

during maturation [54, 55]. Interestingly, in this study, our DGE analysis found some antioxidant genes being downregulated in pig MIX or human MII oocytes, which may expose oocytes to a higher risk of ROS damage.

Numerous studies have reported that supplementation during *in vitro* maturation (IVM) with antioxidants such as vitamin C [56], resveratrol [57], co-enzyme Q10 [58] can increase oocyte quality. The underlying concept is that *in vitro* manipulation of oocytes makes them more susceptible to potential oxidative stress since they lack protective antioxidant mechanisms which are present *in vivo* [59], and components as well as additives in the culture medium including metallic ions and serum albumin can also accelerate ROS generation [60]. Based on this, external antioxidants supplementation enhances the cell's capability to counteract IVM-derived ROS damage. However, the benefits of antioxidant supplementation may depend on the type of antioxidant as vitamin E has been shown to have an adverse impact on bovine oocyte developmental competence [61]. Given the potential influence of IVM process on cell development [62], there might be some genes differentially expressed in MII oocytes in the pig simply because of the *in vitro* maturation conditions. Moreover, gene expression differences between pig and human MII oocytes may not only be derived from species differences, but also from the *in vitro* maturation strategy employed in each species.

Since we did not find any differentially expressed genes between oocytes from the MII stage (visible polar body) and 'damaged' (no visible polar body) oocytes, we assumed that those 'damaged' oocytes still were developmentally competent but were delayed in their development and postponed the extrusion of the first polar body. An alternate explanation is that this process leading to 'damage' may not be dependent upon differentially regulation of gene expression. However, by performing DTU analysis we did find differences in the expression of isoform transcripts. Despite the establishment of several pig transcriptome profiles previously [44, 63, 64], all of them lack a sufficient number of biological replicates (n ≤ 3) for a reliable DTU analysis. We identified in the pig two genes, which had downregulated isoforms without changing the overall gene expression, namely *DTNBP1* (dystrobrevin binding protein 1) and *RAB4B*. *DTNBP1* is associated with normal biogenesis of lysosome-related organelles [65] and has been shown to affect cell cycle progression by interacting with cell cycle-related genes such as cyclins D1/2/3 [66], cyclin B1 (CCNB1), cyclin E1 (CCNE1), cyclin-dependent kinases 1 and 2 (CDK1/2), cell division cycle 20 (CDC20), CDC25A, and CDC25B [67]. *CNB1* and *CDC20* were upregulated in our pig MIX oocytes compared to the GV stage, whereas *CDC25A* and *CDC25B* were downregulated. Additionally, *DTNBP1* was upregulated in bovine embryos that successfully produced live offspring compared with embryos that did not [68]. *RAB4B*, a member of the RAS oncogene family, is involved in protein transport and vesicular trafficking [69]. Previous studies have shown that oocyte maturation is associated with massive macromolecular turnover, which is dependent on intracellular transport and lysosomes [70, 71]. Thus, we propose that reduced utilisation of the predominant translated isoforms of the *DTNBP1* and *RAB4B* genes negatively influences macromolecular turnover, ultimately leading to impaired oocyte maturation. Few studies examined the function of *DTNBP1* and *RAB4B* during oocyte maturation. More work is still needed to test our hypothesis. Furthermore, DTU analysis by comparing MIX versus GV oocytes in the pig dataset or MII versus GV oocytes in the human dataset found many gDTUS undetected by traditional DGE analysis. Those gDTUs were predominantly involved in intracellular transport pathways; ATPase-dependent in pig oocytes and GTPase-dependent in human oocytes. Furthermore, we also found in both, pig and human, that alternative transcription start sites, alternative transcription termination sites and exon skipping were the most frequently-altered splicing events; contributing to isoform switching from full-length isoforms. Moreover, the loss of coding potential caused by isoform switching was observed in pig MIX and human MII oocytes. Such reduced protein-coding

potential in MII oocytes may be associated with impending maternal mRNAs clearance once embryo genome activation occurres after fertilisation [72].

## Conclusion

Our study suggests the failure to progress to MII in vitro may not be regulated at the level of the genome and that many genes are differentially regulated at the isoform level, particular those involved ATPase- or GTPase-dependent intracellular transport.

## Supporting information

**S1 Table. Total list of differentially expressed genes in pig MII oocytes + 'damaged' oocytes (MIX) versus GV oocytes.**
(XLSX)

**S2 Table. Differentially expressed genes associated with oocytes, cell cycle and meiosis in pig MII oocytes + 'damaged' oocytes (MIX) versus GV oocytes.**
(XLSX)

**S3 Table. Differentially expressed genes associated with mitochondria and oxidative phosphorylation in pig MII oocytes + 'damaged' oocytes (MIX) versus GV oocytes.**
(XLSX)

**S4 Table. Differentially expressed genes associated with steroidogenesis in pig MII oocytes + 'damaged' oocytes (MIX) versus GV oocytes.**
(XLSX)

**S5 Table. Differentially expressed genes associated with antioxidant response in pig MII oocytes + 'damaged' oocytes (MIX) versus GV oocytes.**
(XLSX)

**S6 Table. Differentially expressed genes associated with endoplasmic reticulum stress in pig MII oocytes + 'damaged' oocytes (MIX) versus GV oocytes.**
(XLSX)

**S7 Table. Differentially expressed genes associated with ribosomal structure and function in pig MII oocytes + 'damaged' oocytes (MIX) versus GV oocytes.**
(XLSX)

**S8 Table. Differentially expressed genes associated with cytoskeleton and extracellular matrix in pig MII oocytes + 'damaged' oocytes (MIX) versus GV oocytes.**
(XLSX)

**S9 Table. Total list of differentially expressed genes in human MII oocytes versus GV oocytes.**
(XLSX)

**S10 Table. Differentially expressed genes associated with oocytes, cell cycle and cell devisions in human MII oocytes versus GV oocytes.**
(XLSX)

**S11 Table. Differentially expressed genes associated with mitochondria and oxidative phosphorylation in human MII oocytes versus GV oocytes.**
(XLSX)

**S12 Table. Differentially expressed genes associated with steroidogenesis in human MII oocytes versus GV oocytes.**
(XLSX)

**S13 Table. Differentially expressed genes associated with oxidative stress response in human MII oocytes versus GV oocytes.**
(XLSX)

**S14 Table. Differentially expressed genes associated with endoplasmic reticulum stress in human MII oocytes versus GV oocytes.**
(XLSX)

**S15 Table. Differentially expressed genes associated with ribosomal structure and function in human MII oocytes versus GV oocytes.**
(XLSX)

**S16 Table. Differentially expressed genes associated with cytoskeleton and extracellular matrix in human MII oocytes versus GV oocytes.**
(XLSX)

**S17 Table. Upstream regulators for MIX (MII + 'damaged') versus GV stage oocytes in the pig.**
(XLSX)

**S18 Table. Top 25 networks for MIX (MII + 'damaged') versus GV stage oocytes in the pig.**
(XLSX)

**S19 Table. Causal networks for MIX (MII + 'damaged') versus GV stage oocytes in the pig.**
(XLSX)

**S20 Table. Upstream regulators for MII versus GV stage oocytes in the human.**
(XLSX)

**S21 Table. Top 25 networks for MII versus GV stage oocytes in the human.**
(XLSX)

**S22 Table. Causal networks for MII versus GV stage oocytes in the human.**
(XLSX)

**S1 Fig. Morphology of pig oocytes collected for RNAseq.**
(PDF)

**S2 Fig. Heatmap highlighting the distance and gene expression among pig or human oocytes at different developmental stages.**
(PDF)

**S3 Fig. DGE and DTU analysis comparing pig MII oocytes with damaged oocytes.**
(PDF)

**S4 Fig. Alternative splicing mechanisms associated with isoform switches and consequences for protein expression in pig oocytes.**
(PDF)

**S5 Fig. Alternative splicing mechanisms associated with isoform switches and consequences for protein expression in human oocytes.**
(PDF)

## Acknowledgments

We thank Seven Point Pork Abattoir, Port Wakefield, SA for providing the sow ovaries and Mr. Stephen McIlfatrick for the collection of their collection.

## Author Contributions

**Conceptualization:** Katja Hummitzsch, Raymond J. Rodgers.

**Formal analysis:** Feng Tang, Katja Hummitzsch.

**Investigation:** Feng Tang.

**Supervision:** Katja Hummitzsch, Raymond J. Rodgers.

**Writing – original draft:** Feng Tang, Katja Hummitzsch, Raymond J. Rodgers.

**Writing – review & editing:** Feng Tang, Katja Hummitzsch, Raymond J. Rodgers.

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
