## [Decision Letter · Decision Letter 0]

29 Apr 2024

PONE-D-24-10663RNAseq analysis of oocyte maturation from the germinal vesicle stage to metaphase II in pig and humanPLOS ONE

Dear Dr. Rodgers,

Thank you for submitting your manuscript to PLOS ONE. After careful consideration, we feel that it has merit but does not fully meet PLOS ONE’s publication criteria as it currently stands. Therefore, we invite you to submit a revised version of the manuscript that addresses the points raised during the review process.

More experiments are required to support the conclusions.

We look forward to receiving your revised manuscript.

Kind regards,

Meijia Zhang

Academic Editor

PLOS ONE

Journal Requirements:

Reviewers' comments:

Reviewer's Responses to Questions

**Comments to the Author**

1. Is the manuscript technically sound, and do the data support the conclusions?

Reviewer #1: Partly

Reviewer #2: Yes

2. Has the statistical analysis been performed appropriately and rigorously? 

Reviewer #1: Yes

Reviewer #2: Yes

3. Have the authors made all data underlying the findings in their manuscript fully available?

Reviewer #1: Yes

Reviewer #2: Yes

4. Is the manuscript presented in an intelligible fashion and written in standard English?

Reviewer #1: Yes

Reviewer #2: Yes

5. Review Comments to the Author

Reviewer #1: This study employed RNA sequencing to examine the transcriptome profile of oocytes at three major developmental stages including GV, MII and damaged groups of oocytes from the pig, and compared findings with similar public oocyte data from humans. The study suggested the failure to progress to MII may not be regulated at the level of the genome and that many genes are differentially regulated at the isoform level, particular those involved ATPase- or GTPase-dependent intracellular transport.

1. This study focused on transcriptome profile analyses of porcine oocytes from three developmental stages, and there was no direct testing of human samples. So, the title doesn't seem appropriate.

2. All the results in this study are based on RNA sequencing analyses and lack the necessary experimental validation.

Reviewer #2: The authors observed that the transcriptomes in oocytes that failed to progress was similar to those that did in pig. They show that the failure to progress to MII in vitro may not be regulated at the level of the genome and that many genes are differentially regulated at the isoform level, particular those involved ATPase- or GTPase-dependent intracellular transport. This study strengthens the understanding of isoform changes during in vitro maturation from GV to MII oocytes. But the manuscript lacks some clarity in places and the comments below are aimed at trying to improve its clarity and impact.

1. Supplementary Fig 3c, d: Quantitative comparison of all isoforms of DTNBP1 showed that only one isoform was changed while the others were unchanged in pig MII versus damaged MII oocytes. How to understand that DTNBP1 overall transcript levels were unchanged？

2. Is it possible to conduct cellular experiments on RAB4B and DNTBP isoforms in order to reduce the number of damaged oocytes during in vitro maturation?

3. Line 139: As there are 21 sequenced samples in the GSE164371 dataset, please provide the specific GSM number of the human data in this study.

4. Fig 2, Fig4：The critical pathways should be visually emphasized within the corresponding figure by highlighting in red.

5. Fig 2: The labels " up " and " down " should be respectively displayed on 2a and 2b. Similarly, for 2c and 2d, corresponding annotations should be applied to differentiate between the two.

6. Line 314-317: How to analyze the relevant pathway from the gene network displayed in Fig 5? For example, how can gene relationships from network 3 be analyzed to be associated with cell cycle progression? Supplementary Tables 18 also do not seem to directly show the pathways mentioned in the manuscript.

7. Line 319-326: Please indicate which network information in Fig 5 was described in this text.

8. Fig 6: How to analyze the relevant pathway from the gene network displayed in Fig 6? The mentioned genes should be highlighted in these networks.

6. PLOS authors have the option to publish the peer review history of their article (what does this mean?). If published, this will include your full peer review and any attached files.

Reviewer #1: No

Reviewer #2: No

---

## [Author Response · Author response to Decision Letter 0]

20 May 2024

Reviewer 1

COMMENT: This study focused on transcriptome profile analyses of porcine oocytes from three developmental stages, and there was no direct testing of human samples. So, the title doesn't seem appropriate.

REPLY: True! There was no direct testing of human, but we used public available human data to compare with the pig. As such we provide a new perspective. However, to be a bit more accurate we changed the title to ‘The transcriptome profile of oocytes during meiosis: insights from pig and human’.

COMMENT: All the results in this study are based on RNA sequencing analyses and lack the necessary experimental validation.

REPLY: While we understand the importance of experimental validation in confirming the findings of our RNA sequencing analyses, we would like to highlight several points that support the robustness of our results: a) Prior to data analysis, we performed rigorous quality control measures on our RNA sequencing data to ensure data integrity and reliability. This included assessing sequencing depth, read quality, and other metrics to minimise technical artifacts and biases; b) Our RNA sequencing analysis pipeline utilised state-of-the-art statistical methods to identify differentially expressed genes/isoforms and pathways. We employed multiple testing corrections and stringent significance thresholds to minimise false positives and ensure the statistical reliability of our findings. c) The differentially expressed genes/isoforms and pathways identified in our study are biologically plausible and consistent with known biological mechanisms relevant to our research question. Collectively, we respectfully contend that the RNA sequencing analyses presented in our manuscript are robust and sufficiently supported by the evidence presented.

Reviewer 2

COMMENT: Supplementary Fig 3c, d: Quantitative comparison of all isoforms of DTNBP1 showed that only one isoform was changed while the others were unchanged in pig MII versus damaged MII oocytes. How to understand that DTNBP1 overall transcript levels were unchanged?

REPLY: In supplementary Fig 3c, even though there is a significantly higher usage of third isoform in MII oocytes, a lower but not statistically significant usage of first isoform was also found in MII oocytes. Such opposite regulation of isoform usage in MII oocytes makes the overall gene level expression of DTNBP1 vary little and be consistent with that reported for the overall transcript levels being unchanged.

COMMENT: Is it possible to conduct cellular experiments on RAB4B and DNTBP isoforms in order to reduce the number of damaged oocytes during in vitro maturation?

REPLY: This is a good suggestion for future investigation. Its beyond the scope of the existing manuscript but a logical one to follow up on.

COMMENT: Line 139: As there are 21 sequenced samples in the GSE164371 dataset, please provide the specific GSM number of the human data in this study.

REPLY: We now provide the specific GSM number of the human data we used in this study. 

COMMENT: Fig 2, Fig4：The critical pathways should be visually emphasized within the corresponding figure by highlighting in red.

REPLY: Done.

COMMENT: Fig 2: The labels " up " and " down " should be respectively displayed on 2a and 2b. Similarly, for 2c and 2d, corresponding annotations should be applied to differentiate between the two.

REPLY: Done.

COMMENT: Line 314-317: How to analyze the relevant pathway from the gene network displayed in Fig 5? For example, how can gene relationships from network 3 be analyzed to be associated with cell cycle progression? Supplementary Tables 18 also do not seem to directly show the pathways mentioned in the manuscript.

REPLY: The result obtained by IPA software in the supplementary table 18 has shown the top functions of those molecules in the network. For network 3, it is involved in cell progression including cell morphology, molecular transport and RNA trafficking, which are highly correlated with cell cycle progression. Additionally, the high expression of molecules in MII oocytes (red dots) in the network 3 such as NEDD1 and SARNP are potential regulators of cell cycle. NEDD1 has been shown to be required for nucleation of microtubules from the spindle that is important for accurate chromosome segregation during cell cycle [1]. Similarly, SARNP plays a role in cell progression and participates in important transcriptional or translational control of cell growth [2, 3]. Based on above information, we annotated network 3 to be associated with cell cycle progression. Together, we combined the top functions shown by IPA software for the network with biological interpretation of key molecules in the network to do the network annotation. 

COMMENT: Line 319-326: Please indicate which network information in Fig 5 was described in this text.

REPLY: Done.

COMMENT: Fig 6: How to analyze the relevant pathway from the gene network displayed in Fig 6? The mentioned genes should be highlighted in these networks.

REPLY: The same strategy discussed above was used to annotate the gene network in Fig 6. We now highlight the key genes in these networks. 

References:

1. Ma, W., C. Baumann, and M.M. Viveiros, NEDD1 is crucial for meiotic spindle stability and accurate chromosome segregation in mammalian oocytes. Dev Biol, 2010. 339(2): p. 439-50.

2. Fukuda, S., et al., Cloning and characterization of a proliferation-associated cytokine-inducible protein, CIP29. Biochem Biophys Res Commun, 2002. 292(3): p. 593-600.

3. Xie, Y., et al., Structural basis for high-order complex of SARNP and DDX39B to facilitate mRNP assembly. Cell Rep, 2023. 42(8): p. 112988.

---

## [Decision Letter · Decision Letter 1]

7 Jun 2024

The transcriptome profile of oocytes during meiosis: insights from pig and human

PONE-D-24-10663R1

Dear Dr. Rodgers,

We’re pleased to inform you that your manuscript has been judged scientifically suitable for publication and will be formally accepted for publication once it meets all outstanding technical requirements.

Kind regards,

Meijia Zhang

Academic Editor

PLOS ONE

Additional Editor Comments (optional):

Reviewers' comments:

Reviewer's Responses to Questions

**Comments to the Author**

1. If the authors have adequately addressed your comments raised in a previous round of review and you feel that this manuscript is now acceptable for publication, you may indicate that here to bypass the “Comments to the Author” section, enter your conflict of interest statement in the “Confidential to Editor” section, and submit your "Accept" recommendation.

Reviewer #1: All comments have been addressed

Reviewer #2: All comments have been addressed

2. Is the manuscript technically sound, and do the data support the conclusions?

Reviewer #1: Yes

Reviewer #2: Yes

3. Has the statistical analysis been performed appropriately and rigorously? 

Reviewer #1: Yes

Reviewer #2: Yes

4. Have the authors made all data underlying the findings in their manuscript fully available?

Reviewer #1: Yes

Reviewer #2: Yes

5. Is the manuscript presented in an intelligible fashion and written in standard English?

Reviewer #1: Yes

Reviewer #2: Yes

6. Review Comments to the Author

Reviewer #1: (No Response)

Reviewer #2: (No Response)

7. PLOS authors have the option to publish the peer review history of their article (what does this mean?). If published, this will include your full peer review and any attached files.

Reviewer #1: No

Reviewer #2: No

---

## [Editor Report · Acceptance letter]

1 Jul 2024

PONE-D-24-10663R1 

PLOS ONE

Dear Dr. Rodgers, 

I'm pleased to inform you that your manuscript has been deemed suitable for publication in PLOS ONE. Congratulations! Your manuscript is now being handed over to our production team.

Kind regards, 

on behalf of

Dr. Meijia Zhang 

Academic Editor

PLOS ONE